# Effectiveness of the IoT in Regional Energy Transition: The Smart Bin Case Study

**Dimitris Ziouzios** * and **Minas Dasygenis**

Department of Electrical & Computer Engineering, School of Engineering, Campus ZEP Kozani, Active Urban Planning Zone (ZEP), University of Western Macedonia, 50100 Kozani, Greece
* Correspondence: dziouzios@uowm.gr

**Abstract:** As part of the European Green Deal, the EU aims to become climate-neutral and reach net-zero greenhouse gas emissions by 2050. Lignite has long dominated the electricity system of Greece, providing cheap and reliable energy, given the abundant and low-cost domestic resources at the cost of increased emission. In line with its national and international commitments to climate action, Greece needs to urgently transform its energy system and overcome its technological lock-ins, paving the way for a net-zero emission economy by the mid-century. The Internet of Things plays a significant role in this direction, providing with its technologies the protection of the environment and creating new jobs. The smart bins constitute an interesting proposal for areas in the energy transition. This research work reflects the current situation in the region of Western Macedonia and proposes the smart bin project as a part of the solution in the transition to the post-lignite era. For this purpose, survey research has been conducted in the municipalities of Greece on waste management technology.

**Keywords:** recycling; smart bin; smart city; Internet of Things; Western Macedonia; challenges of decarbonization

## 1. Introduction

The global challenge of Reducing Greenhouse Gas (GHG) emissions, recent technological developments and cost reductions of Renewable Energy Sources (RES), the widespread diversification of gas supply sources and the demand for decentralized power generation are leading to a complete and irreversible phase-out from solid fossil fuels, i.e., coal and lignite [1,2]. For the European Union (EU) regions with high dependence of their local economy on the solid fossil fuel industry, the process of decarbonization will require a significant productive diversification in the medium term and, above all, an immediate solution to the problem of thousands of jobs lost in the coming years [3]. The Paris Agreement, the European Green Deal, the Just Transition Mechanism and the Coal Regions in Transition Platform support the aim of just transition, where "no one is left behind", which should be the guiding principle of economic transformations.

In line with its national and international commitments to climate action, Greece needs to urgently transform its energy system and overcome its technological lock-ins, paving the way for a net-zero emission economy by the mid-century. In 2019, the Greek Government has set a goal of withdrawing all lignite plants by 2028, with most units—accounting for more than 80% of current capacity—being withdrawn by 2023 [4]. Given that Western Macedonia has hosted 80 percent of the Greek lignite industry for nearly 70 years, putting the local economy in a position of high reliance on the lignite value chain, Western Macedonia is being urged to not only adapt its production model to the new requirements but also to embark on a comprehensive productive restructuring that will lead to a complete phase-out of lignite activities [5]. In this research work, a reference is made to the lignite industry in Western Macedonia [6], an extensive report is made on the global smart bin market, the smart bin from recyclable materials is briefly presented and the findings of the research carried out in the municipalities of Greece are analyzed. The

overall objective of the paper is to discover the tendency of the municipalities of Greece for the adoption of the electronic solutions provided by the Internet of Things (IoT), specifically concerning smart bins. In addition, there is a greater specialization on the selection of the smart bin from recyclable materials, a project developed by the University of Western Macedonia, and the possible business exploitation is sought in the post-lignite era that the region of Western Macedonia will experience with the decarbonization procedure.

The study proceeds as follows. Section 2 presents the theoretical background and the analytical framework of the research, focusing on the delignification period of Western Macedonia in Greece alongside the smart bin project. Section 3 details the methodology followed for the research. Section 4 outlines the results of the research concerning smart bins in Greece and gives an overview of the local stakeholders and their role in the delignification process. Section 5 discusses the main findings of the analysis, and Section 6 presents the conclusions and future work.

## 2. Theoretical Background and Analytical Framework

### 2.1. Energy Transition in Greece

In recent years, the Greek energy system is characterized by the decreasing consumption of conventional fuels, mainly domestic lignite, which was strategically chosen after the oil crisis of the 1970s as well as out of need for the country's electrification [7]. In September 2019, during the United Nations Climate Action Summit in New York, the Greek Prime Minister Mr. Kiriakos Mitsotakis pledged to phase out all coal-powered electricity production by 2028, with the majority of units—representing over 80% of current installed capacity—being withdrawn by 2023, while only one plant will continue to operate: the Ptolemaida V bloc, which is still under construction in pilot testing and will burn lignite until 2025 or even until 2028 depending on the time it takes to convert it to another fuel supply. According to National Climate Law 4936/2022 article 11, production of electricity from solid fossil fuels will be prohibited from 31 December 2028, and any existing licenses for the production of electricity from solid fossil fuels cease to be valid on this date. The above announcement was captured in the new National Plan for Energy and Climate (NPEC) (https://climate-laws.org/geographies/greece/policies/greece-s-national-energy-and-climate-plan, accessed on 3 February 2023) that was presented in December 2019 following a public consultation and a debate in Greek Parliament. The NPEC is an ambitious plan in accordance with the UN Agenda 2030 and its 17 global Sustainable Development Goals as well as with the recently adopted European Green Deal, setting, in some cases, even higher goals at the national level. Figure 1 presents the participation of lignite in GWh in the balance of energy production in Greece according to the Administrator of Renewable Energy Sources and Guarantees of Origin (DAPEEP SA) at the end of 2021. Even after the sharp increase in the price of natural gas, the production of electricity from lignite remains at the same levels, while the participation of renewable energy sources has increased.

As a result, in 2020, lignite recorded low percentages in the national electricity mix, while natural gas and renewable sources appeared with a steady upward trend and with absolute dominance in the electricity system. The NPEC, among other tools, envisages investments worth 43.8 billion euros in renewable sources; natural gas, electricity transport and distribution networks; financial incentives for the purchase of electric cars; and energy saving by 2030 (http://iobe.gr/docs/economy/en/ECO_Q2_2021_REP_EN.pdf, accessed on 3 February 2023). National financial resources but also European funding will be used, deriving—in particular, by the Just Energy Transition Fund—a new EU financial instrument enhancing an energy transition that is just and socially fair.

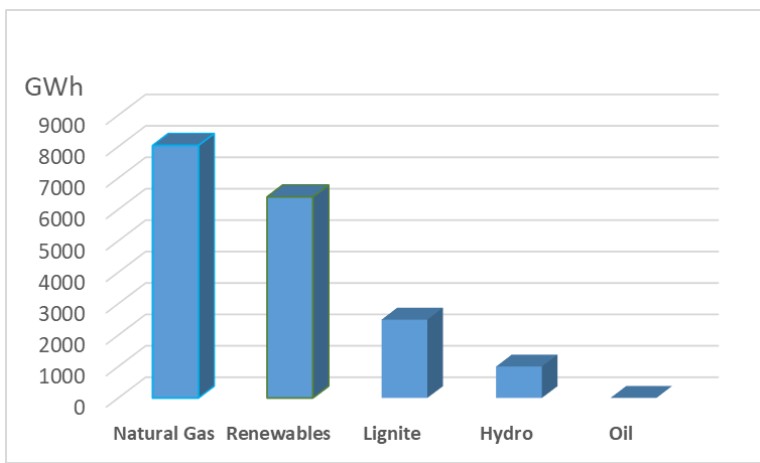

**Figure 1.** Statistics and facts about energy production in Greece.

*2.2. Delignification Process in Western Macedonia in Greece*

Lignite mining in the Ptolemaida area has been going on since the 1920s in small private mines, not in terms of industrial exploitation but in particularly difficult, arduous, persistent and marginally efficient conditions. It is characteristic of that time that the average lignite miner produced about 500 kg of lignite per day, when at the same time in the mines of the German Ruhr, the daily production per employee exceeded 12 t [8]. Since 1956, the production and exploitation of lignite in our region was launched at an intensive pace, fully industrialized, and grew with the involvement of the Public Power Corporation (PPC), with the peak of production in the period 2001–2004; as a result, it was judged as highly effective for our national economy [9]. Since 2002, all indicators such as lignite production, investments in the lignite sector, workforce, etc. show that Western Macedonia has entered a state of delignification. Of course, due to the size of the lignite industry and the inherent inertia that accompanies it, the negative effects began to be experienced in the local community in the decade of 2010. Since 2010, there has been a constant decrease in lignite-fired power plants—the four oldest units stopped operating, a process that accelerated after 2019, triggered by the increased Emissions Trading System (ETS) carbon price, which increased the costs to produce lignite-based electricity combined with policies to promote the use of renewable energy and natural gas. According to the Master Plan (https://www.sdam.gr/sites/default/files/consultation/Master_Plan_Public_Consultation_ENG.pdf, accessed on 3 February 2023) the transition process in Western Macedonia is governed by five basic principles:

- Emphasis on labor-intensive areas to create employment opportunities in local communities;
- Utilization of the inherent advantages of the affected areas;
- Ensuring a quick transition with an emphasis on quick-wins;
- Promoting social and environmental sustainability with an emphasis on sustainable development;
- Integration of modern technology and promotion of innovation.

Based on the above principles, the vision for the "next day" regarding energy transition in Western Macedonia will be based on five pillars of development, as follows in (Figure 2):

- Clean Energy;
- Industry, small industry and trade;
- Smart agricultural production;
- Sustainable tourism;
- Technology and education.

Considering that the region of Western Macedonia has been hosting 80% of the Greek lignite industry for about 70 years, creating the condition of high dependence of the local economy on the lignite value chain, Western Macedonia is called upon not only to adjust its production model to the new requirements but also to proceed immediately

to a comprehensive productive restructuring and a full phase-out of lignite activities. In addition to the NPEC, a National Strategy for Circular Economy has been developed as a horizontal action aiming at the optimal use of resources (energy, water and raw material) in every economic sector. Under a Green Financing Scheme, a series of financing incentives is foreseen for companies investing in circular economy and industrial symbioses, in water reuse after biological treatment, etc. Green innovation concerning sustainable green investments will also be supported. A National Strategy for Adaptation to Climate Change is also being developed, incorporating actions aiming at biodiversity conservation, more effective water resources management, forest management, etc.

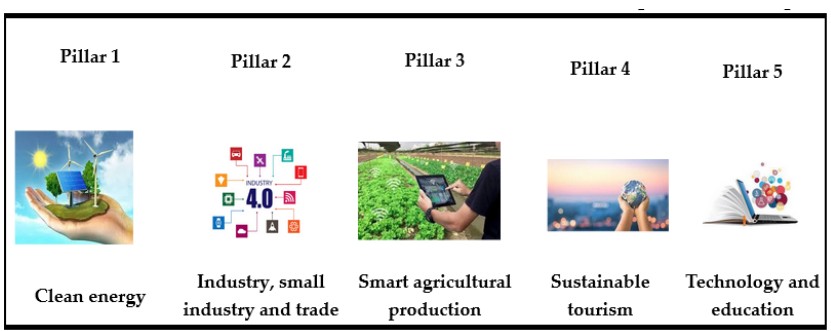

**Figure 2.** Five key pillars of development.

### 2.3. Circular Economy as Part of the Transition Planning

The new National Waste Management Plan (NWMP) pays special attention to recycling and sorting at the source and provides for a separate collection of biowaste for the whole country at the end of 2022, one year earlier than the Community directive. NWMP was developed as Greece's response to the EU Action Plan for the Circular Economy and was endorsed by the Governmental Economic Policy Council in 2018. At the same time, it envisages intensification of the efforts for the separate collection of four streams for recycling, as well as priority in the strengthening of the collection network for recyclable materials. In addition, it sets high recycling targets within the framework of Community obligations (Directive 2018/851; https://www.eea.europa.eu/policy-documents/directive-eu-2018-851-of, accessed on 3 February 2023). The adoption of the principles of Circular Economy and Industrial Coexistence for the overall management and utilization of waste, as secondary raw materials and/or alternative fuels, is also favored. Actions are proposed for the proper and integrated management of the country's agricultural waste, which produces the largest amount of waste (40%) [10] and is managed to date—with individual exceptions—is incinerated at the place of production, resulting in emissions of significant amounts of gaseous pollutants. Also included is the design for the collection and recovery of biodegradable agricultural waste for the purpose of their utilization in the production of by-products (e.g., fertilizer) and/or alternative fuels. Consequently, as shown in Figure 3, there is a significant synergy between the objectives and actions included in the new NWMP, which can have a particularly positive effect on the prospect of entering the smart bin market in the period of 2020–2030 and, in fact, both in the domestic as well as in the commercial sector.

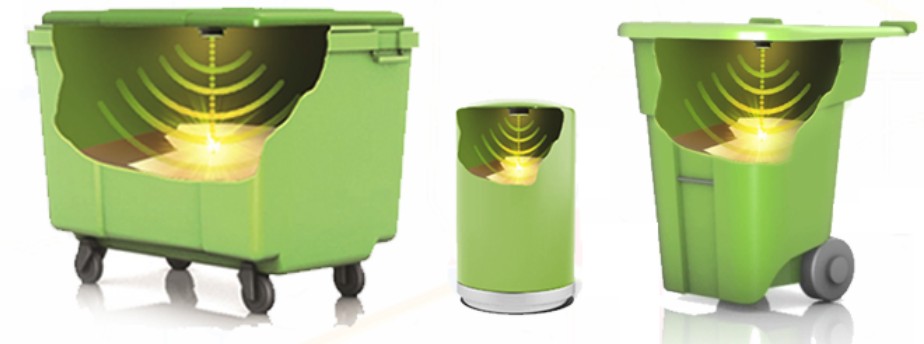

**Figure 3.** Schematic of indicative sizes of smart bins with integrated sensors.

*2.4. The Smart Bin Technology*

The adoption of the new National Waste Management Plan (NWMP) for the period of 2020–2030 (https://www.eea.europa.eu/themes/waste/waste-prevention/countries/greece-waste-prevention-country-profile-2021, (accessed on 3 February 2023), which prefers and supports smart waste management actions, combined with the smart cities philosophy, which is constantly promoted by the European Union, create extremely favorable conditions for the entry of smart bins into the everyday life of the citizen. The ease of automatic disposal of sanitary waste in hospitals and clinics is expected to boost the adoption of automatic refuse bins in the commercial sector [11]. In addition, hotels and restaurants are developing more and more automatic waste containers, especially for public health reasons, as part of their efforts to offer attractive services to customers. The main idea behind the adoption of smart bin technology is primarily related to the hygiene of public spaces. The problem we face in a modern city is empirically known by the citizens, where the neighborhood glass or paper bins are regularly overflowing and often their collection is delayed. As a result, citizens dump their rubbish next to bins, rubbish is scattered on the street and lost in the recycling process, and additional costs of cleaning the urban area and garbage collection are required. The consequences are obvious: degradation of public space and additional costs for taxpayers. Smart waste management with smart bins could, if not eliminate it altogether, significantly reduce the above problem. Obviously, waste management companies have the ability to calculate their routes and collection intervals according to experience and seasonal factors, such as celebrations, gatherings or important local events. However, edge technology could help to optimize the cleaning process and the immediate collection of waste. This practice would not only save significant financial resources but also help to avoid overfilling bins. Finally, every kilo of glass or paper that is not lost for recycling means extra money for the municipality.

Smart bins are designed to identify different types of waste. They consist of IoT-enabled sensors that act as real-time indicators to determine if the bins are full or not and help adjust the waste collection schedule accordingly. Smart bins are environmentally friendly, as they significantly reduce the need for misguided collection routes, resulting in reduced carbon dioxide ($CO_2$) emissions and associated greenhouse gases. The use of the compression system maximizes the capacity of the bin, significantly increasing the recycling rates, while the smart bins are standardized in such a way that they can be emptied using the existing collection equipment. According to the Danish Environmental Protection Agency, collecting a cup of coffee thrown in the street costs the municipality services about 2 euros to collect and drive into the trash bin. On the other hand, according to the same office, one third of Copenhagen's citizens' complaints to cleaning services are consistently related to overfilled bins [?]. Therefore, smart waste management with a focus on smart bins not only significantly improves the management costs for municipalities but also creates cleanliness and hygiene conditions in the urban environment and is a constant requirement on the part of citizens. A variety of smart bin systems are currently available on the global market, many of which are still in a pilot phase.

According to an extensive study conducted in 2017 in Denmark regarding the future of the smart bin market, the following interesting findings arise [8]: Low cost sensor: The total direct replacement of existing trash bins with smart bins is not a viable and attractive option for most public cleaning services. Therefore, the solution of adding sensors to existing bins is preferred. The sensor system must be quite inexpensive, so that if the bin is damaged or stolen it does not mean big financial expense or replacement. Based on the interviews, a cost of EUR 80 per bin, for the addition of sensors, would be perfectly acceptable on the part of municipal authorities. A further problem is the fact that waste collection bins are often burned or vandalized. This possibility makes municipalities reluctant to invest a lot of money in the smart bin market, especially in large urban centers. Construction Simplicity: Complex, high-tech solutions have long been unattractive on the municipal side. Municipalities usually choose simple but effective solutions, unlike, of course, service businesses. A very advanced and expensive device in a bin is not a selection criterion for a municipality; moreover, it will make it difficult to repair the technical services of the municipality. According to our research, municipalities want smart bins but not technologically complex ones. Open/transparent system: Great emphasis is placed on open data, open codes and compatibility. Municipalities are looking for solutions that can work together and where they have the freedom to change between different systems without great difficulties. This could, for example, mean that a municipality can choose other sensors; use the sensors on another platform; or even further, develop its own communication systems. In addition, the study highlighted a number of other findings:

- Many potential customers (municipalities) already have a large number of conventional bins; so, a high price for new smart bin systems is definitely discouraging.
- It is much easier for a municipality to receive funding from the Central Government to hire cleaning staff (job creation) rather than funding to implement an advanced waste management system.
- While most smart bin solutions ensure durability for about a decade, customers are skeptical about these claims at 60%.
- All public entities are looking for open-source solutions so they can easily build on it, modify or add new solutions. Most existing solutions are proprietary.
- Many public bodies, even at the director level, are not sufficiently informed about the technological philosophy of smart bins.

*2.5. The Global Market of Smart Bin Technology*

The global market of smart waste management is expected to reach USD 4.5 billion by 2025 [12], as shown in Figure 4. This market is projected to grow at a rate of 7% from 2019 to 2027 in revenue, mainly due to the adoption of the potential Internet of Things (IoT) and the technological philosophy of smart cities [13]. During the projected period (2019–2025), North America is expected to become the leading smart waste collection market, followed by Europe and Asia-Pacific. The growing government initiatives, strict environmental regulations and large-scale investments with the emergence of IoT have brought about a real breakthrough, reducing the operational costs of these technologies. Moreover, the increasing adoption of IoT technology, which links a range of smart sensors and devices to monitor and automate city waste management functions, positively affects the market for intelligent waste collection around the world. Regarding geographical distribution, North America is expected to dominate the smart waste collection market, as the US has a significant market share and is expected to expand by 8.1% annually in the period of 2019–2027. In addition, the smart waste collection market in Europe and the Middle East is also projected to increase during the forecast period.

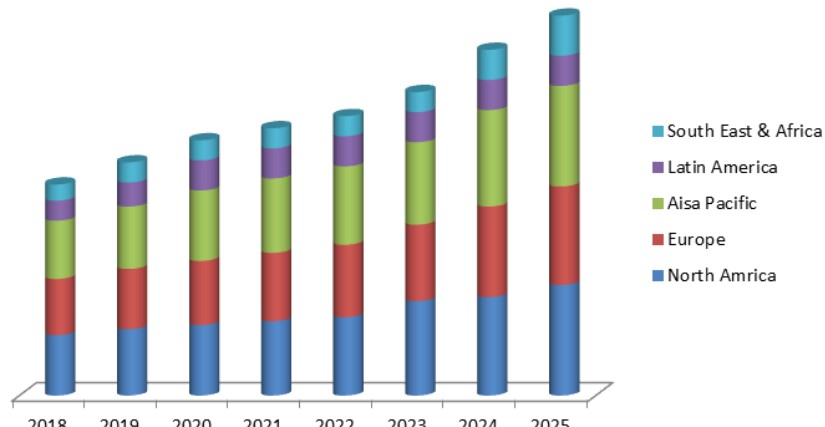

**Figure 4.** Geographic market distribution of smart bins with apparent growth around the world.

Particularly for smart bins, which are a key parameter of the smart waste management philosophy [14], the market is expected to grow at a rate of 64.1% to reach USD 5.42 billion in 2025, according to a recent survey by Frost and Sullivan [13]. The global market for smart waste bins is currently estimated at USD 278.8 million. According to Frost and Sullivan's study (https://www.researchandmarkets.com/reports/4856435/growth-opportunities-in-the-global-internet-ofproduct--toc, accessed on 3 February 2023), to gain a competitive advantage in the smart bin market, Frost and Sullivan needs to make the most of the growth opportunities that present themselves: Delivering value-added services such as cleaning and maintenance, on-site waste audits and working with other providers to develop a complete package of smart waste infrastructure.

- Expanding into areas that are rapidly urbanizing and generating large amounts of waste.
- Partner with providers for efficient design, installation and distribution of smart bins.
- Developing a wide range of products, especially to enhance the entire value chain; for example, partnerships with Big Databases and image recognition technologies.
- Launch new platforms that can manage data created from any connected smart bin device and transfer information about the use and performance of bins.

The waste management chain includes individual stages such as collection, transport, disposal and recycling. Especially, the stage involving collection and disposal shows the highest operating costs leading to the growing adoption of intelligent waste management. In the smart collection department, the emergence of the internet has revolutionized and effectively dealt with operational costs for waste management companies. Companies that offer smart practices for collecting waste focus primarily on three solutions: intelligent tracking, path optimization and data collection. By developing sensors, network infrastructure and data visualization platforms, waste management companies have been able to generate actionable information and make informed decisions. As a result, municipalities in some cities in the United States, the United Arab Emirates, the United Kingdom, etc., in cooperation with innovative waste management (such as Enevo, Smartbin, Bigbelly, etc.) save around 30% of the waste collection costs [10]. After all, the smart bin market in North America is as follows (https://www.grandviewresearch.com/industry-analysis/north-america-automatic-touchless-garbage-bin-market, accessed on 3 February 2023):

- The growing trend for Smart Cities, especially in Canada and the USA, is leading to an increase in the market for smart cities, both in the commercial and domestic sectors.
- A smart city leverages digital technology for better resource utilization and fewer broadcasts. This means more intelligent urban transport networks, upgraded water supply and waste disposal facilities, and more effective ways of lighting and heating buildings. It also means a more interactive and sensitive city administration, safer public spaces and meeting the needs of older people.

- About 22% of cities in the United States and Canada have already implemented smart bin strategies, compared with just 7% of cities globally.
- Due to government initiatives promoting sustainability, zero waste by 2020 and the penetration of smart city initiatives across the high-urban concentration region, North America is expected to represent the largest share in the smart waste management market.

Also remarkable is the growing cooperation between waste management companies, material recycling companies and companies that develop software and automation. Recently, predominantly due to the pandemic of COVID-19, there has been a growing demand for hands-free containers in all public services and in the commercial sector. A development is expected to significantly accelerate the demand for smart bins across the economy. Around 150 companies are active worldwide offering solutions in the smart bin sector, including five flagship start-ups (https://www.startus-insights.com/innovators-guide/5-top-smart-bin-solutions-impacting-smart-cities/, accessed on 3 February 2023), as shown in Figure 5, offering integrated solutions and, above all, developing highly innovative practices that are expected to give technological advancement to the smart bin market.

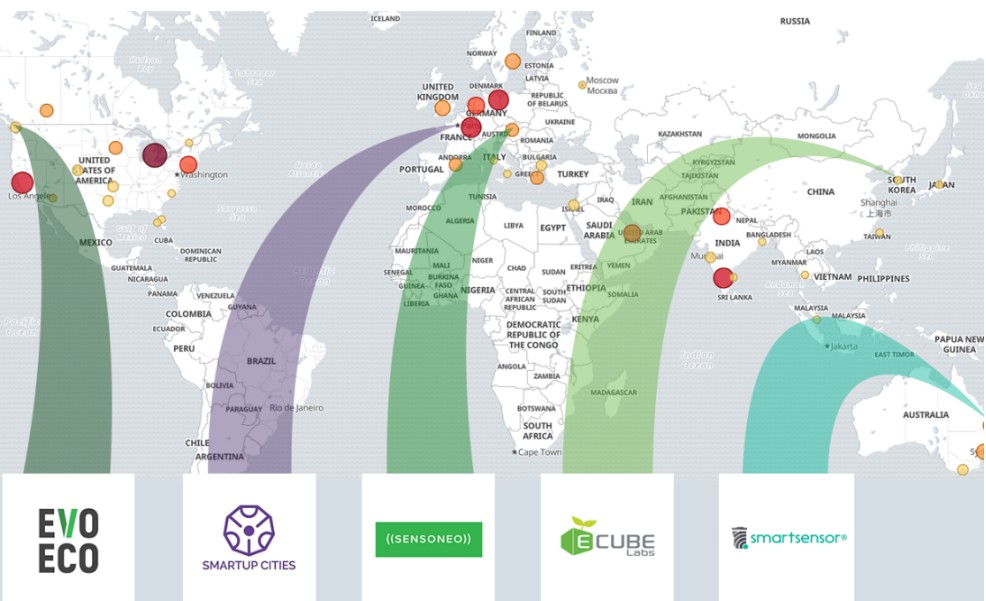

**Figure 5.** Global diaspora of businesses active in smart bin production.

The current waste management process starts with the creation of waste, which is disposed of in waste bins near the point of its creation. The waste is then collected from the garbage dumps of municipalities or private companies at the prescribed times and transferred to temporary collection centers. Garbage at collection centers is then sent either for recycling or to landfills. This process partially solves the waste problem while creating other problems such as the following:

- Some garbage bins are overfilled, while others are not met by garbage collection time;
- Overloaded garbage cans create unsanitary conditions;
- Unoptimized truck routes lead to excessive fuel use and environmental pollution;
- Collected garbage complicates sorting at the recycling facility.

An example of a modern intelligent waste management system will include a trash-connected sensor that measures the fill level and a communication system that transfers data to the Cloud [15]. The data are processed in the Cloud; therefore, the routes of the collection trucks are optimized. Smart waste management contributes to overall waste recycling efficiency, provides the opportunity to optimize the route for utilities and helps reduce fuel traffic and consumption. In conclusion, it seems that the smart bin solution concerns entrepreneurship at the international level and can bring tangible results both

economically and environmentally. Due to this, market research has been conducted in Greece regarding the knowledge of the smart bin and the possible utilization of this technology.

### 2.6. Our Smart Bin Implementation

It took 3 years of research for our team to develope an environmentally friendly smart bin [16,17], as shown in Figure 6, which is constructed of environmentally friendly materials and uses the Internet of Things (IoT) to optimize garbage truck routes and cut fuel use. The main principle is to employ a glue derived from polyester recycling, which has been granted a patent by the Greek Industrial Property Organization with the number 1004906. This special adhesive is made by dissolving polystyrene $(C_8H_8)n$ in an acetone solution $(CH_3COCH_3)$.

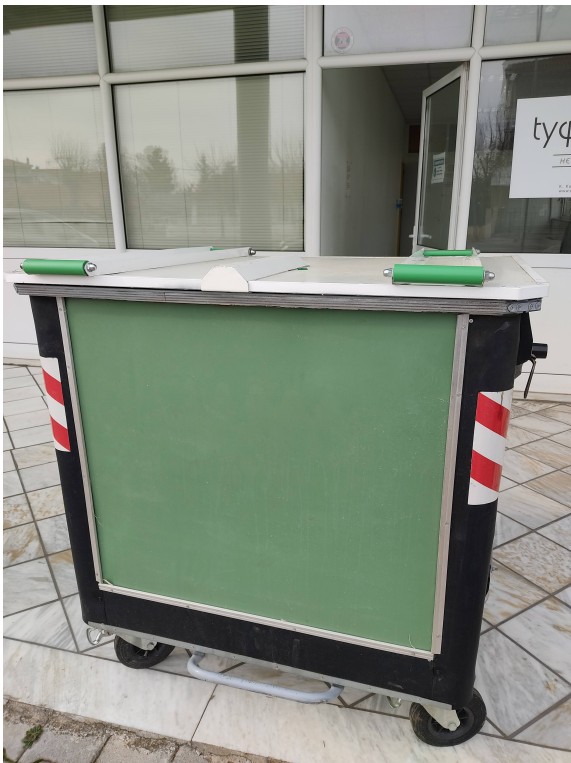

**Figure 6.** The proposed smart bin.

The advantages of this technology include the easy recycling of polystyrene, as well as the ease with which it can be manufactured and its good welding qualities. Both of these features help to keep production costs down. This product may be used to glue tiny or big pieces of wood, metal, glass, leather or paper and serve as the foundation for composite product manufacture. The glue is viscous and has a fluidity of 1800 $mm^2/s$ at 20 °C. As the cardboard is made from recycled paper and contains a considerable quantity of lead (Pb), its usage as an inert material in bin construction has a major beneficial environmental impact. The electronic system is based on the microcontroller Arduino and has a typical power consumption circa 200 mW. For this project, the microcontroller is ideal due to the low power consumption and price. The electronic system has a sonar for calculating the fill percentage of the bin, a temperature/humidity sensor and a LoRa Antenna for sending the measurements to a Lora Gateway and finally to the Information System (IS). The IS is used by the person in charge of garbage collection so that they have real-time information about the condition of the bins. The main benefits of this waste bin are as follows:

- Lighter than conventional bins;
- More durable in the case of vandalism;
- Resistant to burning;
- Smart, utilizing the Internet of Things.

The goal is to produce the bin in the region of Western Macedonia by combining the low carbon footprint and the Internet of Things. This will have multiple benefits for the area. Above all, it will contribute to the reduction of carbon dioxide; to the better management of municipal solid waste; and to the economy of the region, both to specialized personnel and to the workforce that already has the area such as unskilled workers, drivers, etc.

## 3. Methodology

This case study was carried out as a survey questionnaire [18]. The questionnaire was designed to provide initial information about the knowledge on smart bin technology and its benefits for stakeholders in Greece; at the same time, it reflects the current situation of municipal waste management in the country. The managing authority of municipal solid waste in Greece is the respective municipality and, for this reason, the 332 municipalities of the entire country were selected, regardless of population or other characteristics. The questions were refined a number of times and the questionnaire was short, at 13 questions, using appropriate language and terms to be perceived. The questionnaire was designed on Google Forms, in Greek language, and a link was sent to the stakeholders by e-mail. Telephone communication with the municipalities was also needed in order to complete the questionnaire, since we are in a time of the COVID-19 pandemic, personal contact was impossible and the time available to municipal officials was minimal. As a result, the questions were few in order to better reflect the current situation regarding municipal waste management and smart bin technology. The answers given were grouped so that they could be better captured in percentages and graphs. We based this survey research on the stakeholders' theory analysis framework. In an organization, a stakeholder is someone who can affect or is influenced by the success of the organization's goals [19]. There are plenty of research areas that utilize the stakeholder theory, such as energy development, social services, mining operations, the tourism industry, project governance and e-government problems [20–23]. The stakeholder theory is suitable and can be used to understand the needs of stakeholders involved based on the method of the waste management system used. Based on the stakeholder theory, Theodoulidis et al. [24] proposed the use of two models that explicitly investigate the relationship between stakeholder management, expressed as corporate social responsibility (CSR) activities, firm strategy and corporate financial performance (CFP). Furthermore, various metrics and procedures, such as measures of interest and influence, have been created to recognize and analyze these stakeholders [25]. Vittola et al. [26] investigated the impact of national culture, an external determinant, from a stakeholder theory perspective. In some cases, the stakeholders are categorized so that the result will be more targeted, e.g., into primary and secondary stakeholders [27]. The values of the following three indicators were used to identify stakeholders:

- Influence: How much is stakeholders' influence based on their status, talents, demeanor and bargaining rights? It was anticipated that if stakeholders had effective means to influence policy, their impact would be significant. If they did not, their effect was assumed to be negligible.
- Involvement: This indicates the proportion of all stakeholders who are participating in the waste management system, as well as aspects such as resources, information and technology. Involvement was considerable, as evidenced by a significant percentage, and the inclusion of additional elements was modest, as evidenced by a small fraction and the contribution of fewer factors.
- Interest: This indicates how much the process of waste management affected the stakeholders' interests. The stakeholders' interest was judged to be high if the policy had a direct impact on them; if it did not, it was considered that their interest was low.

In Greece, the municipals are in charge of the whole procedure of waste management. They were chosen to take part in this survey. After the answers to the questionnaire, a brief description of the smart bin from recyclable materials was made, which was developed by the University of Western Macedonia, which is briefly described in Section 2.6. The aim of this research is to investigate the knowledge of experts on the technology of smart bins and

then draw useful conclusions about whether the smart bin from recyclable materials can contribute to the business gap that will be created by delignification in Greece, mainly in the region of Western Macedonia.

## 4. Analysis and Findings

In order to record the primary preparedness of the Market at the national level in Greece, regarding the perspective of the smart bins, the research team drafted and sent a targeted questionnaire, as presented in Annex I—Questionnaire of this report, to the 332 municipalities of the country, of which 103 municipalities replied (39%). The municipalities that responded come from all regions of Greece and 53.8% are medium-sized municipalities (from 10,000–60,000 inhabitants), 23.1% are large (over 60,000 inhabitants) and the remaining 23.1% are small municipalities (under 10,000 inhabitants). The processing of the replies showed interesting findings.

The above findings show that municipalities are interested in adopting smart and innovative waste management practices, advocating that investments in smart bins are sustainable and efficient. In any case, the targeted information of municipalities on the benefits of the use of smart bins as well as the implementation of pilot demonstration projects will greatly accelerate the use of smart bins on a wide scale, creating an ever-growing market. The interest of municipalities on the exploitation of the Internet of Things (IoT), which also includes smart bins as a priority, is a great opportunity to increase the market in the coming years. Municipalities are also prepared to pilot smart bins.

Figure 7 shows that 26.9% of municipalities know quite a lot about smart bin technology, 30.8% know enough and 42.3% know almost nothing about it. So, it is obvious that more than 60% of the representatives have some knowledge about smart bin technology. In Figure 8, we can see that almost all municipalities, more than 90%, are planning to adopt innovative technologies regarding smart waste management systems. Figure 9 shows that 38.5% are interested in the immediate replacement of conventional bins with smart ones and 57.7% are planning it in the medium term. Another interesting finding is that 96.2% of municipalities would not be opposed to paying the increased purchase price of smart bins compared with conventional ones, considering the benefits that will result from their use, as shown in Figure 10. Finally, 96.2% of municipalities are planning to transform their cities to smart directly or in the medium term plan, as shown in Figure 11.

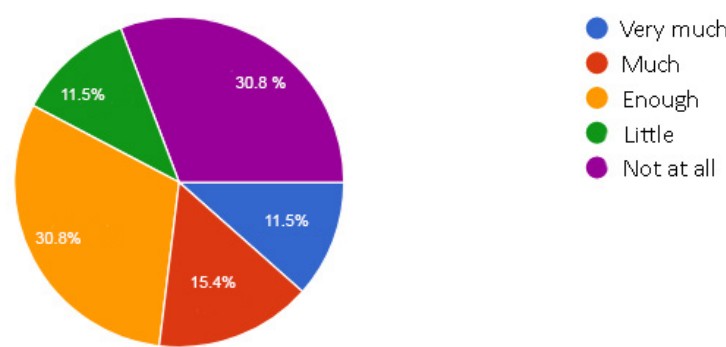

**Figure 7.** A total 26.9%know quite a lot about smart bin technology, 30.8% know enough and 42.3% know almost nothing about it.

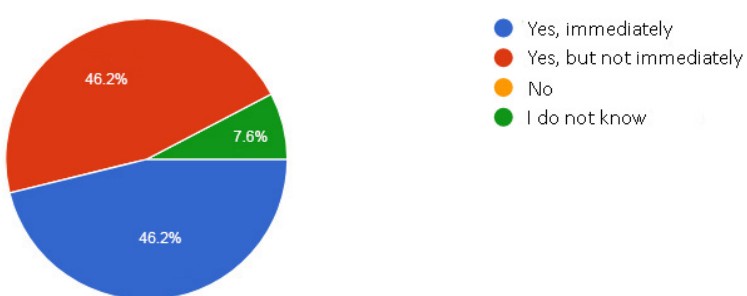

**Figure 8.** Almost all municipalities are planning to adopt innovative technologies regarding smart waste management systems.

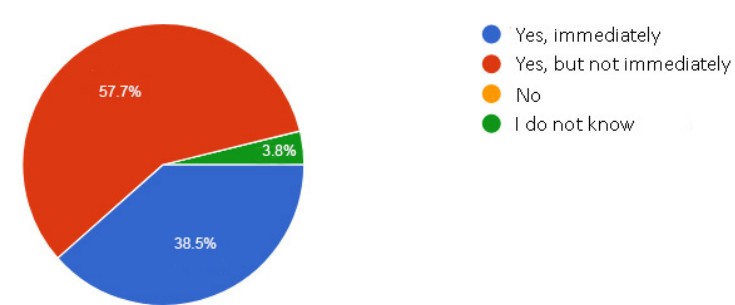

**Figure 9.** A total 38.5% are interested in the immediate replacement of conventional bins with smart ones and 57.7% are planning it in the medium term.

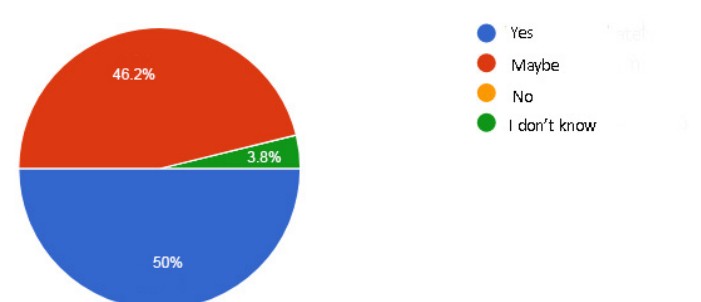

**Figure 10.** A total 96.2% of municipalities would not be against paying the increased purchase price of smart bins compared with conventional ones, considering the benefits that will result from its use.

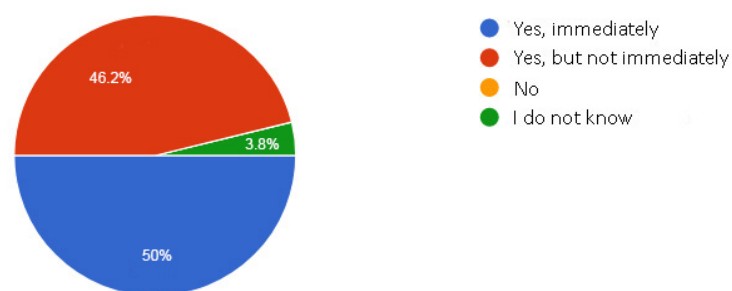

**Figure 11.** A total 96.2% of municipalities want to transform directly or in the medium term plan to smart city practices (smart cities).

Through this research, it is easy to understand that municipalities across the Greek territory positively view the possibilities offered to them by the use of new technologies

and, in particular, the use of the smart bin in order to better address waste collection and management. Even municipalities that do not have the necessary knowledge or information about the possibilities offered by the smart bin are prepared to be informed.

## 5. Discussion

Greece needs to minimize GHG emissions and reorganize the local economy by phasing out lignite operations while implementing investment and generating employment in productive sectors and industries to drive a fair transition in Western Macedonia. Western Macedonia's economic reform is not a subject of sectoral policy; it must be properly integrated into practically all governmental programs. The EU is aggressively pursuing ambitious emission-reduction plans with the goal of being climate neutral by the mid-century and phasing out fossil fuels, particularly coal and lignite. Given that this is a complicated, multi-level and multi-annual process, the planning of the transition strategy in Western Macedonia should be coordinated in a clear and effective manner. Policies should identify the essential variables that promote the region's socioeconomic development while guaranteeing a just transition that leaves no one behind. In this context, we proposed as part of the solution the construction of the smart bin in Western Macedonia. The local economy's strong reliance on lignite mining has generated barriers and structural vulnerabilities in the region's socioeconomic growth. The National Energy and Climate Strategy, the PPC's business plan and the EU's ambitious environmental rules all state that lignite will be phased out in the prosperous years. Therefore, there is an urgent need for effective policy interventions and actions that will support Western Macedonia to create its own path towards a just transition; socioeconomic development; regional resilience; and a prosperous, diversified economy and society. The current study has yielded valuable insights that may be applied by other carbon-intensive regions in Europe and beyond to help them transition to a low-emission, resilient and sustainable economic paradigm. In particular, the case study demonstrated the application of the smart bin project in the direction of diversifying the local economy towards more productive sectors proposed by the EU and the Greek government. With investments in new technologies such as solar PV, power storage, green hydrogen and IoT, Western Macedonia can play a vital role in the transformation of Greece's energy system. To connect development goals with existing human capital and infrastructure, a framework to assist regional development might be devised. By boosting the region's attractiveness, the active promotion of environmental sustainability and inclusive growth may be leveraged to attract new people. Policy tools are needed to support significant collaboration between the energy sector, industrial activity, the primary sector, and research and development. As technology is already available and Greece's scientific human capital is a valuable asset, if these factors are brought together in a coherent forward-looking strategy, they can help Western Macedonia achieve sustainable development by creating new high-value-added jobs while also protecting the local environment, air quality and human health.

## 6. Conclusions and Future Work

Greece and especially the region of Western Macedonia faces serious socioeconomic problems in the transition of the delignification era. The lignite process is well-established and employs the largest percentage of workers in the wider area. The cooperation of local authorities, communities and organizations could assist with new initiatives to ensure the effective orientation of the regional economy towards productive, sustainable and green activities. Through the questionnaire research conducted in Greece, it is clear that the municipalities are oriented towards the use and utilization of the Internet of Things in order to improve the quality of life of the inhabitants and to improve the general management of the issues. The managers of the municipalities aim to invest money in cutting-edge technologies, which are oriented in this direction. The transition to renewable energy sources and the release from lignite create major problems in areas in transition, such as that of Western Macedonia. In order to solve the problem in order to have the best

possible result, a series of parallel actions must be taken to create new jobs. The smart bin project can be a part of supporting the economy in Western Macedonia by utilizing the Internet of Things and emphasizing the circular economy. In this paper, the current situation prevailing in Greece and specifically in Western Macedonia was presented and the possibility of the smart bin from recyclable materials was examined as to whether it can be part of the solution with a socioeconomic impact in Greece. As discovered through the research conducted, Greece is ready to take the next step in the digital age, aiming to invest money and utilize the Internet of Things. Specifically, it was studied whether the municipalities of Greece are willing to replace the conventional waste collection bins with smart bins. The vast majority agree to replace them. Further, the smart bin developed by our research team was proposed, which has two additional characteristics of being fireproof and greater durability. It is obvious that smart bins have a great buying interest and could positively contribute to the transition of delignification in Western Macedonia. Further research should be performed in the wider region of the Balkans and Europe in order to investigate the demand for smart bins from other countries. With this in mind, the smart bin could be made from recyclable materials in Western Macedonia in order to absorb labor and not further reduce the population due to decarbonization.

**Author Contributions:** Methodology, D.Z.; formal analysis, D.Z.; investigation, D.Z.; writing—original draft, D.Z.; writing—review and editing, D.Z.; supervision, M.D. All authors have read and agreed to the published version of the manuscript.

**Funding:** This research has been co-financed by the European Regional Development Fund of the European Union and Greek National funds through the operational Program Competitiveness, Entrepreneurship and Innovation, under the call RESEARCH-CREATE-INNOVATE (project code: T1EDK-01864).

**Institutional Review Board Statement:** The study was conducted in accordance with the Declaration of Helsinki, because no humans or animals were used in the study. It was not necessary to be approved by the Institutional Review Board (or Ethics Committee) of UOWM.

**Informed Consent Statement:** Informed consent was obtained from all subjects involved in the study.

**Data Availability Statement:** The research data is not publicly available.

**Conflicts of Interest:** The authors declare no conflict of interest.

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
