# Peer review of "Effectiveness of the IoT in Regional Energy Transition: The Smart Bin Case Study"

_recycling, doi:10.3390/recycling8010028_

Round 1

Reviewer 1 Report

Der Author

The methodology and analysis and findings are well-written. I think your survey about smart bins and your developed smart bin is very interesting. But your introduction needs some improvement because you talk a lot about lignite fire plants and the lignite industry. But as a reader of your publication, you must read till page 9 to understand the connection between lignite and smart waste bins. I would suggest shortening the introduction and investing more in the analysis and finding part. Your findings need more space, you should describe them more and reference them in the text. All consider your source referencing and citations should be improved because a lot of facts are not referenced.

Here, you please find some further comments:

Line

Comment

16

Unnecessary t after Greece

13-14

Nearly the same sentence as in Lines 38-39

Line 13-14: Lignite has long dominated the electricity system of Greece, providing cheap and reliable energy, given the abundant and low-cost domestic resources at the cost of increased emission.

Line 38-39: Lignite has long dominated the electricity system of Greece, providing cheap and reliable energy, given the abundant and low-cost domestic resources.

74-75

Is 2025, right?

the Ptolemaida V bloc, which is still under construction and will burn lignite until 2028 or even until 2025

80-81

Source and Data from which year?

Fig. 1 how much the participation of lignite in the balance of energy production in Greece has decreased.

138

157

Different abbreviation for National Waste Management Plan

Line 138 NWMP Line 157 NRP

Figure 1

Need for a source reference and year of the presented data

68-70

Need for a source reference for the decreasing consumption of conventional fuels

71

How was the Greek Prime Minister?

72—73

Need for a source reference for the phase-out for the coal powerplants and the 80% of current installed capacity

78-79

NPEC need a source reference to be accessible to others

80-86

The presented facts are not presented in Figure 1

You talk about participation in Line 80 but show GWh in Figure 1

86-87

A source of 43.8 billion euros is needed.

93-97

 unnecessary information

99

No explanation for PPC

100-101

Source for the peak of production in the period 2001-2004

101

Which indicators

104-107

Source needed for constant decrease lignite fires power plants and the increased Emissions Trading System (ETS) carbon price

108

Source for the Master Plan

138

The new national Waste Management Plan (NWMP)

Need for a source and a year

Also for Lines 138-142

144

Need for a source reference for (Directive 2018/851)

148

Source for largest amount of waste agriculture (40%)

Figure 3

Need for an image source reference

157

Need for a source reference new National Waste Management Plan (NRP) for the period 2020-2030

161-166

Source for the statement

166-168

Tone mismatch:

The problem is known to all of us: our neighborhood glass or paper bins have overflowed, but according to the collection plan, the waste management company will return the next day. The result:

194

A variety of smart bin systems are currently available on the global market, many of which are still in a pilot phase, as table 1 shows

There is no table 1.

209-210

Source for the statement: Municipalities want smart bins, but not technologically complex ones.

216-226

Source for these statements

Figure 4

Need an additional axis

Not clear what is shown

228-229

The global market of smart waste management is expected to reach USD 4.5 billion by 2027 [9], as shown on Fig. 4.

Figure 4 only shows data from 2018 to 2025 and no data of 2027

It isn't clear what is shown in this Figure (no y-axis)

228

244

246

Three different abbreviations for US Dollars

Line 228 USD; Line 244 US$; Line 246 $278.8 US

229-232

Source for the statement: This market is projected to grow at a rate of 7% from 2019 to 2027 in revenue, mainly due to the adoption of the potential Internet of Thinks (IoT) and the technological philosophy of smart cities. During the projected period (2019-2027), North America is expected to become the leading smart waste collection market, followed by Europe and Asia-Pacific.

238-242

Source for the statements: Regarding geographical distribution, North America is expected to dominate the smart waste collection market, as the US has a significant market share and is expected to expand by 8.1% annually in the period 2019-2027. In addition, the smart waste collection market in Europe and the Middle East is also projected to increase during the forecast period.

245-247

Source reference needed for Frost & Sullivan

246

$278.8 US

Million or billion?

258-265

Need for Sources for these paragraphs.

271-283

Source for these bullet points

293-297

Need for Sources for these paragraphs.

303-310

Need for Sources for these paragraphs.

325-326

Format of the units: 20 oC and 1800 mm2/s

329

Power consumption isn’t measured in mA

352

Unnecessary 1 in municipalities1

Figure 7-11

There is no reference to these figures in the text

Also, the sharpness and the used font size aren't always the same

424

Waste Macedonia is mentioned a couple of times, why is it now with an abbreviation

434

Unexplained abbreviation PPC

Author Response

Thank you for your remarks

Author Response

Thank you so much

Round 2

Reviewer 1 Report

Dear Author,

thank you for considering my comments und suggestions.

Best wishes.